# CATransU-Net: Cross-attention TransU-Net for field rice pest detection

**Xuwei Lu** [1]*, **Yunlong Zhang**[1], **Congqi Zhang**[2]

**1** Henan Agricultural Information Data Intelligent Engineering Research Center, SIAS University, Zhengzhou, China, **2** School of Software Engineering, Chengdu University of Technology, Chengdu, China

* zswwyy125@163.com

## Abstract

Accurate detection of rice pests in field is a key problem in field pest control. U-Net can effectively extract local image features, and Transformer is good at dealing with long-distance dependencies. A Cross-Attention TransU-Net (CATransU-Net) model is constructed for paddy pest detection by combining U-Net and Transformer. It consists of encoder, decoder, dual Transformer-attention module (DTA) and cross-attention skip-connection (CASC), where dilated residual Inception (DRI) in encoder is adopted to extract the multiscale features, DTA is added into the bottleneck of the model to efficiently learn nonlocal interactions between encoder features, and CASC instead of skip-connection between encoder/decoder is designed to model the multi-resolution feature representation. Compared with U-Net and Transformer, CATransU-Net can extract multiscale features through DRI and DTA, and enhance feature representation to generate high-resolution insect images through CASC and decoder. The experimental results on the large-scale multiclass IP102 and AgriPest benchmark datasets verify that CATransU-Net is effective for rice pest extraction with precision of 93.51%, about 2% more than other methods, especially 9.36% more than U-Net. The proposed method can be applied to the field rice pest detection system. Code is available at https://github.com/chenchenchen23123121da/CATransU-Net.

## 1 Introduction

In the world, rice, as an important food crop, plays an irreplaceable role in maintaining food security. Rice planting area and total output rank first in the world. Rice is the main crop in China, having a stable planting area of 30 million square hectometers, and an annual output of more than 2 trillion tons [1]. However, in the process of rice planting, all kinds of pests are happening to seriously threaten the yield and quality of rice, such as Brown Plant Hopper, Green Plant Hopper, Stem Borer, Gallmidge, Army worm, Rice leaf roler, Rice Gundhi Bug, Mole Cricket and Rice Hispa, as shown in Fig 1 [2]. The Food and Agriculture Organization (FAO) reports that rice pests cause annual losses of 20% up to 40% percent of global rice production [3].

**Data availability statement:** Data are available from the https://github.com/xpwu95/IP102.

**Funding:** This paper was supported by the Science and Technology Project of Henan Province in 2024 (Nos. 242102210021, 242102110377). The funders had no role in study design, data collection and analysis, decision to publish, or preparation of the manuscript.

**Competing interests:** No authors have competing interests.

The development of efficient and accurate pest detection methods and technologies in paddy fields is of great significance for taking timely control measures and reducing pest losses [4]. Traditional rice pest monitoring relies on farmer experience, which cannot meet the needs of large-scale and diversified pest monitoring in smart agriculture.

From Fig 1, it is known that it is difficult to accurately and completely identify the detecting pests from pest images due to the irregular shapes, different sizes and complex background. Recently, many deep learning (DL) models have been presented with outstanding results in various fields [5,6], including the detection, classification and identification of crop pests and diseases [7]. However, DL-based pest detection remains important and challenging, due to the various crop pests with changeable sizes and shapes, including striped pests, circular pests and colored pests, as shown in Fig 1A and B, as well as many small pests are not obvious and very similar to the field background, as shown in Fig 1C and D.

The simple DL models have some shortcomings in various pest detection, which may lead to missing and false detection [8,9]. Hybrid DL models combining residual, multiscale convolution and Transformer are increasingly active research with great potential for development [10]. Inspired by the hybrid DL models, such as residual cross-spatial attention-guided Inception U-Net model (RCA-IUnet) [11], TransUNet [12], UNetFormer [13], Residual U-Net and Transformer (RUNet) [14], CNN-Transformer [15], Modified U-Net(MU-Net) [16], Lightweight Multi-Scale Dilated U-Net (LMSDU-Net) [17] and TinySegformer [18], a Cross-Attention TransU-Net (CATransU-Net) is constructed for rice pest detection. Since deconvolution can only enlarge the image but not restore it, to reduce feature information loss, cross-attention skip-connection (CASC) instead of skip-connection of U-Net is added by clipping the left-hand down-sampling maps to the same size and then directly concatenating it back together. Through CASC and dual Transformer-attention module (DTA), the low-level location information is integrated with the deep semantic information. Compared with the existing pest detection methods, it makes use of the merits of U-Net, residual U-Net (RU-Net), multiscale U-Net (MSU-Net), attention U-Net (AU-Net), and TransUNet for rice pest detection. Its main contributions are summarized in three folds:

- Dilated residual Inception (DRI) is adopted to extract the multiscale features, making the model robust to irregular pests.

- CASC instead of skip-connection can capture context semantic information between encoder-decoder features, by integrating semantic information into low-level features and spatial resolution into high-level features.

- DTA at the bottleneck of the model can integrate both local and global contextual information, providing a comprehensive understanding of both local and global contexts.

The rest of paper is arranged as follows. The related work is simply discussed in Section 2. Section 3 describes the architecture of CATransU-Net and its components in detail. Experiments and result analysis are presented in Section 4. Finally, Section 5 concludes the study and presents the future work.

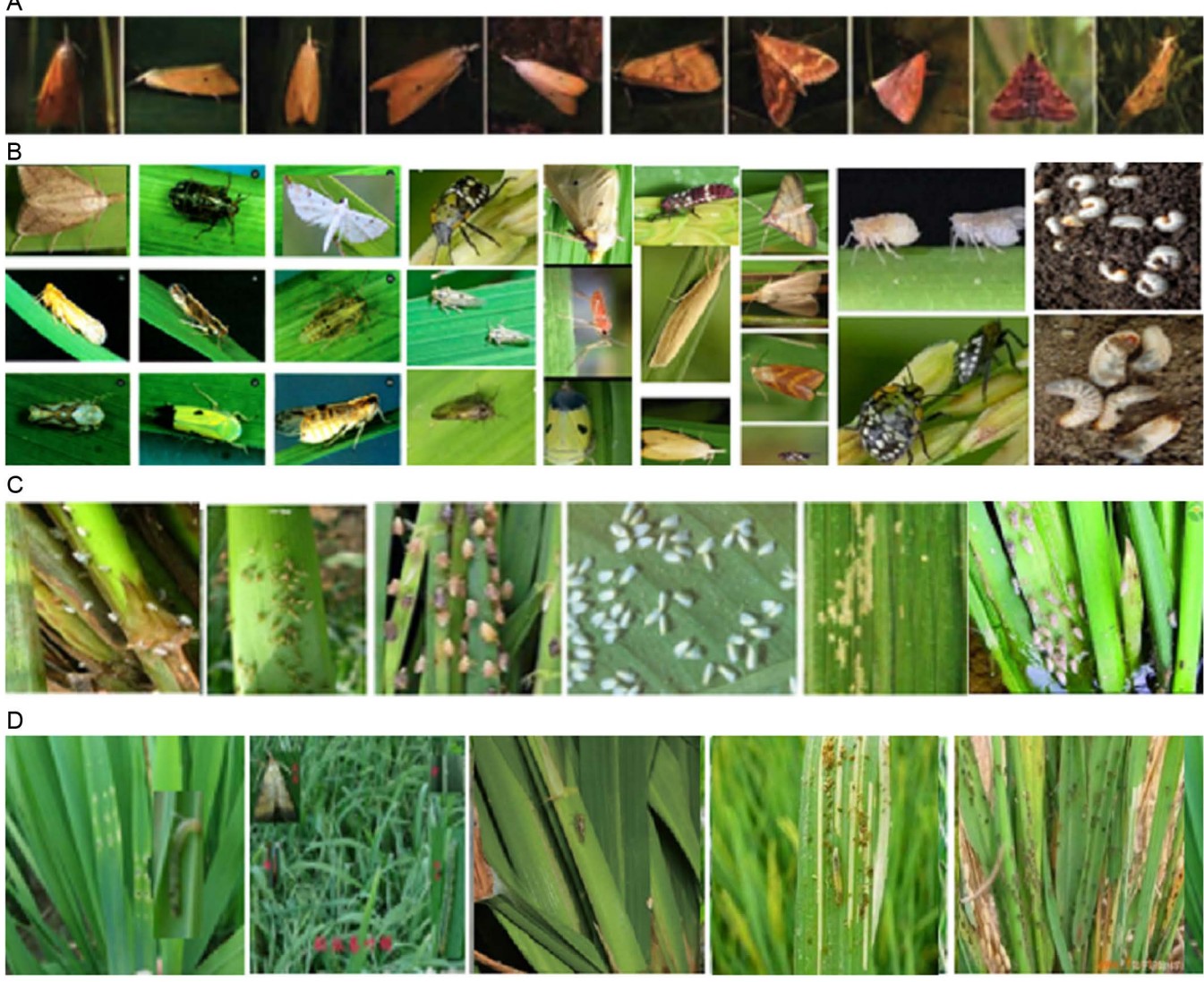

**Fig 1. Rice pest image examples.** (A) A pest with different shapes and sizes. (B) Various rice pests with different shapes and sizes in the IP102 dataset. (C) Tiny rice pests in the field in the AgriPest dataset. (D) Not obvious rice pests.

## 2 Related work

With the continuous development of computer vision and its application [19,20], many DL models have been widely applied to crop pest detection field [21]. These models are divided into simple DL models, multiscale convolution models, attention mechanism models and Transformer models.

### 2.1 Simple DL models

DL can gradually learn more discriminant features such as lines, edges, corners, simple shapes, and various shapes from complex images. Wu et al. [22] reviewed the DL-based pest identification methods, introduced the application and development of different DL models, analyzed the latest research progress in this field, compared their respective performance and characteristics, and summarized the existing pest image datasets. Liu et al. [23] summarized the recent research on

crop diseases and insect pest detection based on DL, including classification, detection and segmentation according to different network structures, and concluded the advantages and disadvantages of each method. Chithambarathanu et al. [24] introduced many ML and DL models and their applications, and introduced the research progress of DL models such as convolutional neural network (CNN), long short-term memory (LSTM) and their variants in the field of crop pest detection and recognition. These models enable automatic monitoring of crop pests and diseases over large areas, reducing human error and workload. Ali et al. [25] reviewed the research on DL-based crop pest detection from four aspects: classification network, detection network, segmentation network and practical application, and summarized the advantages and disadvantages of recent crop pest detection methods based on DL, and put forward the future trend in the field. Guo et al. [26] described the mainstream DL models applied to pest detection and identification, introduced several public datasets of pest images, outlined the challenges faced by current DL-based pest detection and recognition algorithms, and pointed out future research directions in the field.

The above approaches and models tend to extract a lot of irrelevant features from pest images, but fail to take full advantage of the spatial detail information of the pest image, which easily lead to overfitting problem and poor generalization in complex environments.

## 2.2 Residual, multiscale, and attention convolution models

To improve the performance of pest detection in the field, many multiscale, residual, and attention convolution models have been presented. Chen et al. [27] adopted pre-trained MobileNet-V2 as the backbone network, and proposed an attention-embedded lightweight network for crop pest recognition. Xu et al. [28] constructed a multi-scale convolutional capsule network for crop pest identification. It can learn multi-scale convolution features from different pest images for pest image recognition. Zhang et al. [29] proposed a lightweight pest detection method named AgriPest-YOLO. In the model, the coordination and local attention mechanisms are employed to obtain richer pest features and reduce backgrounds and noise interference. To improve the performance of multi-scale pest detection under complex background, Wang et al. [30] constructed a three-scale attentional dilated CNN model for multi-scale pest detection. It can focus on crop pests and diseases, overcome information loss, accelerate network training, and reduce the influence of background factors. Later, they constructed a dilated multi-scale AU-Net (DMSAU-Net) model for crop insect pest detection [31]. The results on the public dataset IP102 validate that it is superior to the other models. Yang et al. [32] proposed a crop pest detection model named YOLOv5s-pest. In the model, a hybrid spatial pyramid pooling module is used to capture multi-scale information, and a convolutional block attention module (CBAM) is used to improve detection precision. Lv et al. [33] proposed a lightweight multi-scale feature extraction model for pest detection based on ResNet and attention mechanism, and verified it on the constructed rice pest and disease dataset. Vhatkar et al. [34] proposed a DL-based pest detection method using an adaptive single-lens detector. In the method, the multi-scale expansion attention network is used to classify pests, and the model parameters is fine-tuned to improve the accuracy and efficiency of the model.

The above pest detection methods employ different improved optimization approaches to improve the detection accuracy. However, they are difficult to capture the local and global features at the same time, and easy to lose image details because of the essential pooling and up-sample operations.

## 2.3 Transformer models

Transformer is powerful in modeling the global context [35]. It consists of 6 encoder-decoder modules, each encoder module is composed of a multi-head SA layer and a feedforward neural network layer, the input of the attention layer K and V is the output of the corresponding encoder module. Recently, Transformer has been widely introduced into DL-based image segmentation and detection tasks and achieved remarkable success. It uses a multi-head self-attention (MHSA) to extract global context feature and has the ability of parallel training. Dai et al. [36] proposed a pest detection in plants by a YOLOv5m, where Transformer is added into YOLOv5m to capture the global features, increase the receptive field,

and weighted concatenation is used to enhance the feature fusion capability of the network. Zhang et al. [37] proposed a multimodal fine-grained Transformer (MMFGT) model for pest identification. The model consists of self-supervised learning to expand Transformer performance, contrast learning to extract target features, and multimodal information fusion to improve the performance of pest identification. To improve detection accuracy under challenging conditions, Bai et al. [38] introduced an adaptive optimizer and a lightweight pest detection method by combining Transformer and super-resolution sampling approach. Fang et al. [39] proposed a large-scale multi-class field pest classification method based on CNN-Transformer, consisting of a hybrid convolution Transformer encoder, a self-supervised mask autoencoder and a fine-grained visual classification module. Mehta et al. [40] constricted a Lightweight Transformer namely MobileViT by using convolutions and transformers in a way that the resultant MobileViT block has convolution-like properties while simultaneously allowing for global processing. This modeling capability allows us to design shallow and narrow MobileViT models, which in turn are light-weight.

The above pest detection methods utilize Transformer to enhance the global feature representation, but they ignore the deviation between the spatial and semantic features, resulting in low detection accuracy. Due to the characteristics of paddy pests and the complexity of field background, and paddy pests have high intra-class difference and inter-class similarity, the rice pest detection in field natural environment is still challenging research. As for the problem of paddy insect pest detection, a Cross-Attention TransU-Net (CATransU-Net) model is introduced to address the detection problem of irregular changes in the size and shape of paddy insect pests.

## 3 Method

CATransU-Net is a modified MSU-Net model by introducing a dual Transformer-attention (DTA) module and a cross-attention skip-connection (CASC) module into MSU-Net to improve the performance of paddy pest detection in the field. The structure of CATransU-Net is shown in Fig 2.

From Fig 2, it is seen that CATransU-Net is based on U-Net architecture, consisting of 4 main components: encoder, decoder, CASC and DTA. Their structures are shown in Fig 3.

The purpose of the pest detection task is to predict the semantic label mapping. Assume the input pest image is $X \in R^{H \times W \times C}$, where $H \times W$ is the size of $X$, and $C$ is the number of channels. The four main components are described as follows.

1) Encoder. It has 4 residual dilated Inception (RDI) modules, each is composed of 4 parallel dilated 3×3 convolutions with dilated rates of 1, 3, 5 and 7 with a residual connection followed by $Conv_{1\times1}$＋BN operations, concatenating, a $Conv_{1\times1}$＋BN, and a ReLU activation, as shown in Fig 3A. It uses RDI to carry out convolution and pooling operations, and obtain low-resolution feature map, reflecting pixel level semantic information. In RDI, the 1×1 convolution module

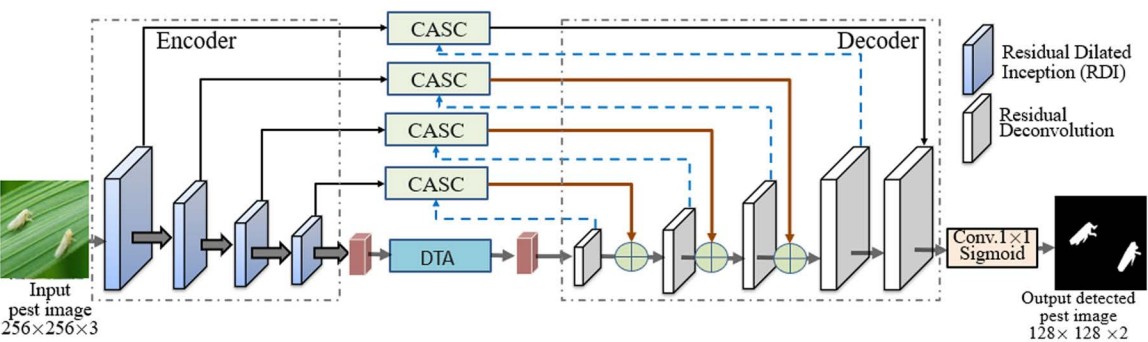

**Fig 2. The structure of CATransU-Net.**

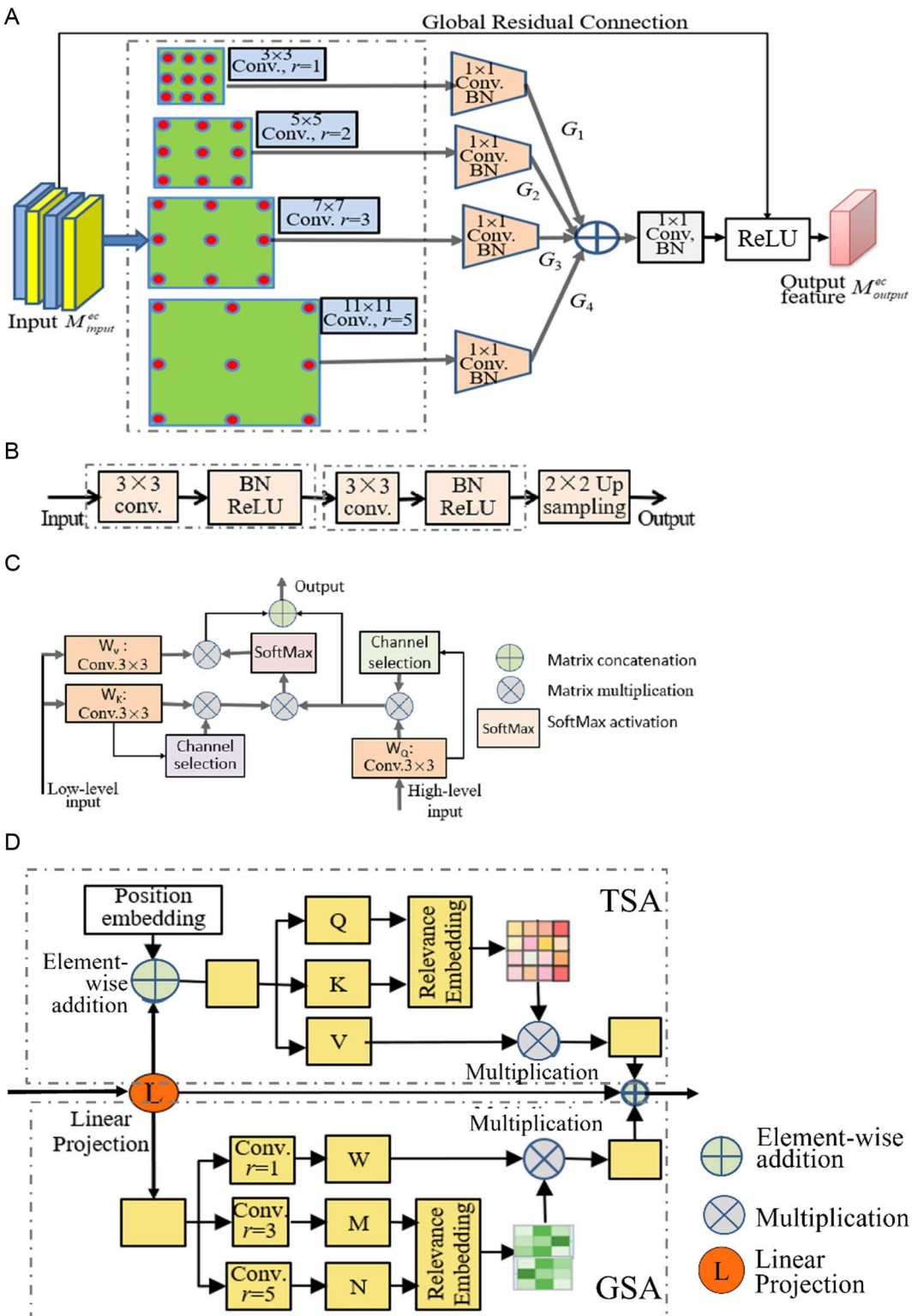

**Fig 3. The main components of CATransU-Net.** (A) Residual dilated Inception (RDI) module in encoder. (B) Deconvolution in decoder. (C) CASC. (D) DTA.

can effectively reduce the number of feature channels. Different from classical convolution, RDI can enlarge the receptive field and extract multiscale features. The output from 4 RDIs are passed to DTA.

2) Decoder. It is a residual deconvolution module for feature extraction and fusion, including two $Conv_{3\times3}$ + ReLU + BN and a 2×2 up-sampling, as shown in Fig 3B. The output of the first decoder is input into the second decoder, and so on. In this way, the feature map is gradually transformed as meaningful features closer to the target. The output of the last decoder is input into a 1 × 1 convolution layer, followed by the sigmoid to generate the binary detection image.

3) CASC. It is a context feature fusion module to connect the low-level input from the encoder and the high-level input from decoder. It can enhance the semantic features of encoder and decoder with attention weight. Its structure is shown in Fig 3C. CASC aims to restore high-level key features to high resolution, present pixel-level semantic information in the original image, and obtain the final pest image with pixel-level semantic information. The low-level input is reshaped to generate matrices $K$ and $V$, the high-level input is reshaped into a matrix $Q$, and channel selection is used to get the key channel of $K$ and $Q$, the SoftMax is used to carried out the multiscale dot-product operation between the transposed versions of $Q$ and $K$, generate a cross-attention map to represent the global similarity of a given element semantically and spatially, the map is multiplied with $V$ matrix to obtain an aggregation of contextual concern weights, and the reshaped low-level and high-level features is concatenated to obtain the final output of CASC. The key channel is selected as follows,

$$
\begin{aligned}
P_c &= \frac{1}{h\times w}\sum_{m=1}^{h}\sum_{n=1}^{w}F_c(m,n) \\
A_w &= sigmoid(W\cdot P_c) \\
\tilde{F} &= A_w\cdot F
\end{aligned}
\tag{1}
$$

where $P_c \in R^c$, $F$ and $\tilde{F} \in R^{c\times h\times w}$, $F_c$ is the feature of the $c$-th channel of the input $F$, $W$ is the weight, $A_w$ is the weight of all channel features, $\tilde{F}$ is the critical channel feature.

4) DTA. It consists of Transformer Self-attention (TSA) and Global Spatial Attention (GSA). Their structures are shown in Fig 3D. TSA and GSA are incorporated into CATransU-Net to efficiently learn the multi-scale non-local interactions between encoder features. They are introduced as follows.

(1) TSA aims to obtain contextual information from the global representation subspace by multi-head attention mechanisms. Suppose the encoder features $F_{en} \in R^{c\times h\times w}$. It is embedded into three matrices, $Q, K$ and $V \in R^{c\times(h\times w)}$, defined as follows,

$$
Q = F_{en}\cdot W_q, K = F_{en}\cdot W_k, V = F_{en}\cdot W_v
\tag{2}
$$

where $W_q$, $W_k$ and $W_v$ are three weighted matrices corresponding to three linear projections.

*TSA* is calculated as,

$$
TSA(Q,K,V) = SoftMax(QK^T/\sqrt{d_k})V
\tag{3}
$$

where $\sqrt{d_k}$ is the dimension of the sequence of $F_{en}$.

Reshaping the feature maps by Eq (3) to generate the output of TSA as $F_{TSA} \in R^{c\times h\times w}$.

(2) GSA aims to selectively gather the global context features into the learned features, encode the context location features $F_{en} \in R^{c \times h \times w}$ of the encoder into the local features, and generate two feature maps by two different types of $F_p^c \in R^{c \times h \times w}$ and $F_p^{c'} \in R^{c' \times h \times w}(c' = c/8)$, where $F_p^{c'}$ is reshaped and transformed into two feature maps $Map_1$ and $Map_2$, and $F_p^c$ is transposed into $W$. $Map_1$ and $Map_2$ are matrix multiplicated with SoftMax normalization to obtain position attention maps $M_{pos} \in R^{(h \times w)(h \times w)}$, defined as:

$$M_{posij} = \exp(Map1_i Map2_j) / \sum_{i,j=1}^{n} \exp(Map1_i Map2_j)$$

(4)

where $M_{posij}$ is the weight of the $i$-th position on the $j$-th position, and $n = h \times w$ is the number of pixels.

$W$ is multiplied with $B$, and the resulting feature at each position is calculated as:

$$GSA_i(Map1, Map2, W) = \sum_{j=1}^{h \times w} (W_j M_{posij})$$

(5)

Reshaping the feature maps by Eq (5) to generate the output of GSA as $F_{GSA} \in R^{c \times h \times w}$.

From the above analysis, the weighted feature map of DTA is calculated as follows,

$$F_{DTA} = \gamma_1 F_{DTA} + \gamma_2 F_{GSA} + F_{en}$$

(6)

where $\gamma_1, \gamma_2$ are two scale-parameters to control the importance of SA and spatial attention maps, respectively.

In experiments, $\gamma_1, \gamma_2$ are initialized as 0, and gradually increased with the important features.

Model training. In model training, Adam (Adaptive Moment Estimation) optimizer is adopted to train the model, and the balanced binary cross-entropy loss function is adopted to train CATransU-Net in an end-to-end manner, calculated as follows:

$$Loss = \frac{1}{n_{pix}} \sum_{p=1}^{n_{pix}} [gt_p \cdot \log(sig(pred_p)) + (1 - gt_p) \cdot \log(1 - \log(sig(pred_p)))]$$

(7)

where $n_{pix}$ is the number of pixels of an input image, $gt_p$ is the ground truth, $pred_p$ is predicted masks, and $sig(.)$ is Sigmoid activation function.

## 4 Experiments and discussion

In this Section, a large number of experiments are performed on the rice pest image subset of the IP102 dataset (https://github.com/xpwu95/IP102) to evaluate the performance of the proposed method, and compare with three baselines: U-Net, multi-scale U-Net (MSU-Net) [41] and attention U-Net (AU-Net) [19], and five state-of-the-art pest detection methods: TransUnet [12], DMSAU-Net [31], U-Net with hybrid DL mechanism (HDLU-Net) [7], Swin Transformer [42] and TinySegformer [18]. TransUnet is a hybrid architecture that utilizes detailed high-resolution spatial features of CNN and global context information encoded by Transformer to achieve precise target positioning. DMSAU-Net is a modified U-Net,

including dilated Inception and attention mechanism. HDLU-Net is a is a modified U-Net by combining CNN and Gated Recurrent Unit (GRU). Unlike common ConvNet models, Swin Transformer introduces shifted window-based self-attention mechanisms to efficiently model the local and global dependencies in pest images. TinySegformer is a hybrid DL model by combining the advantages of Transformer, self-attention mechanism and neural networks.

## 4.1 Dataset

**4.1.1 Dataset 1.** IP102 is a public crop pest image dataset. It contains more than 75,000 images of 102 crop pests, belonging to 5 kinds of field pests (Rice, Corn, Wheat, Beet, Alfalfa) and 3 kinds of economic crop pests (Vitis, Citrus, Mango), where 19,000 images in JPG format are annotated with bounding box for pest detection and recognition. The dataset has 8515 rice pest images belonging to 14 kinds of rice pests, which can be download through (https://github.com/xpwu95/IP102) [43], and their indexes and names can be obtained from https://github.com/xpwu95/IP102/blob/master/classes.txt. Fig 4 and Table 1 show the detailed information and examples of rice pest image subsets, belonging to a total of 14 kinds of pests, with different shapes and sizes, as shown in Fig 4A and B. Fig 4C and Table 1 show the image distribution of rice pest image subset.

**4.1.2 Dataset 2.** AgriPest is a public dataset(https://github.com/liuliu66/AgriPest), which is employed to validate the proposed model. The data contains 49.7 dry images of 14 pests of four crops in a field environment. All the images were manually labeled by agricultural experts using pest positioning bounding boxes up to 264.7K. The dataset defines, categories, and establishes a series of detailed and comprehensive domain-specific sub-datasets. Its first category contains two typical challenges: pest detection and pest population counting. Subsequently, the dataset categories four types of the validation subsets of AgriPest dense distribution, sparse distribution, illumination variations, and background clutter, which are common in practical pest monitoring applications.

## 4.2 Data preprocessing

From Fig 4C and Table 1, it is seen that the subset of rice pest images is very unbalanced, where the number of pest images of Rice leaf roller has 1115 images and the number of pest images of Grain spreader thrips has only 173 images. To mitigate bias in IP102, which may suffer from class imbalance, leading to poor model generalization, we use data augmentation to increase the number of pest images in small-class, as shown in Fig 4D.

To consider the tradeoff between detection performance and computational complexity in model training, each pest image is randomly cropped to 640×640 in a unified manner, and is annotated using free software called labelling. To compare with the existing methods, the pest images are resized to 256×256 for training and test in a unified manner due to the limitation of GPU memory.

## 4.3 Experiment set

All the above models are implemented by DL framework PyTorch 1.13.0, Ubuntu18.04, NVIDIA RTX3090 with 24G video memory, Intel(R) Core(TM) i9-10900K. Five-fold cross-validation (FFCV) is adopted to implement experiments, optimizer is Adam to train the model with learning rate of 0.001, weight decay of 0.0001, and momentum of 0.9, mini-batch size of 32, the weights in the backbone are initialized by the "Kaiming" method, and the total number of iterations is set as 3000. In the absence of improvement, the learning rate is reduced by a factor of 0.1. Adam(Adaptive momentum) is selected as optimizer because it can effectively overcome the overfitting problem in model training.

Precision (*Prec*) and Recall (*Rec*) are often applied to evaluate model accuracy and completeness in detecting positive samples, calculated as follows,

$$\text{Prec} = \frac{|SF \cap GF|}{|SF|}, \text{Rec} = \frac{|SF \cap GF|}{|GF|}$$

(8)

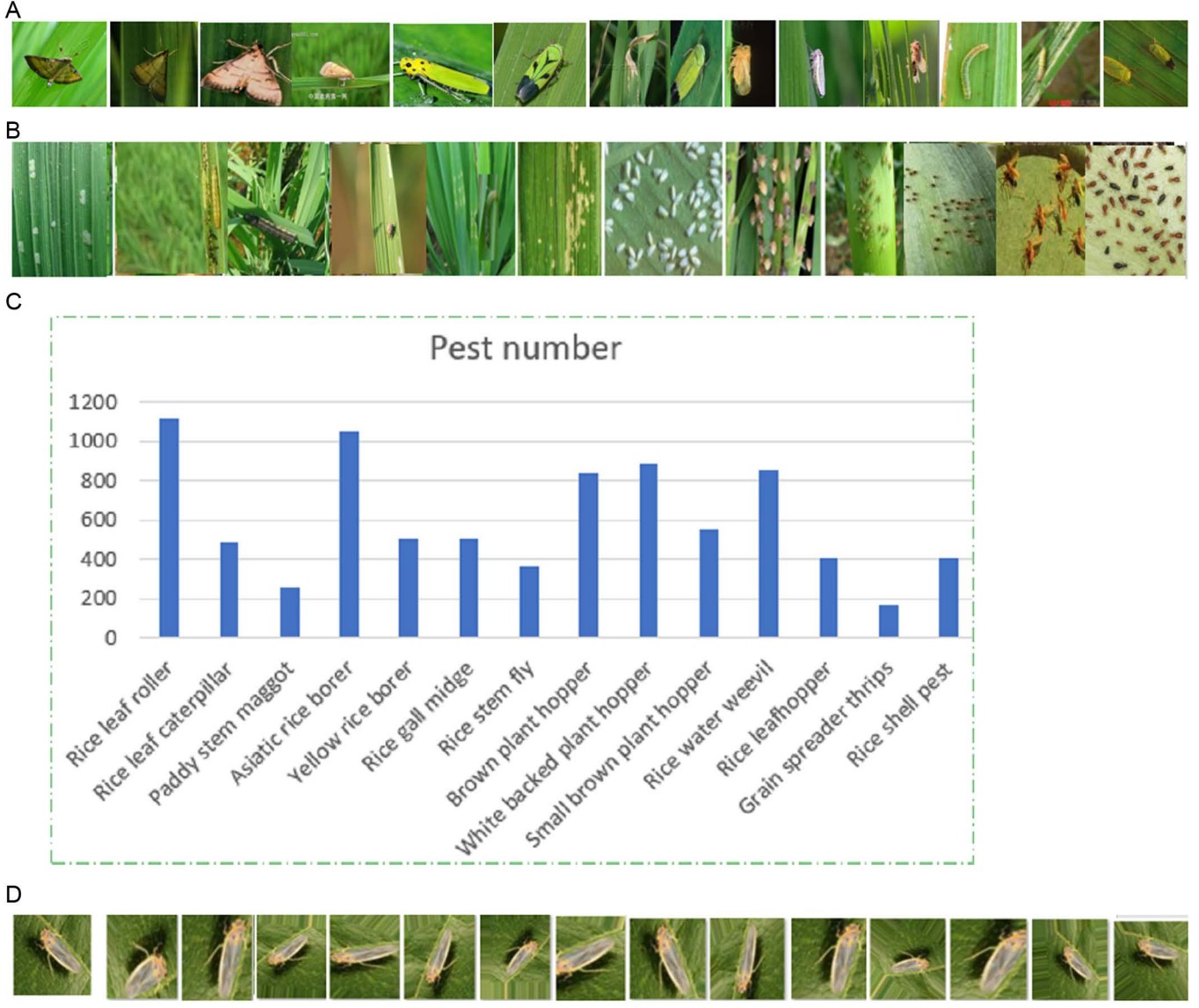

**Fig 4. Rice pest image examples and Class distribution.** (A) Simple pest images. (B) Complex pest images. (C) The number of pest images of different categories. (D) 14 augmented images of the first pest image.

where *SF* is the set of foreground-pixels, *GF* is the set of pixels belonging to the foreground in the ground truth.

*Prec* and *Rec* curves are drawn with vertical axis to denote the precision score and horizontal axis to denote the recall score. Mean Average Precision (mAP) is adopted to evaluate the model performance across categories. It is a comprehensive measure of precision and recall by considering *Prec*, *Rec* and different *mAP* thresholds. It is especially suitable for multi-class pest detection task, calculated as follows,

$$mAP = \frac{1}{M} \sum_{i=1}^{M} AP(i)$$

(9)

**Table 1. The image distribution of rice pest image subset.**

| Class No. | Numbering interval | Pest name | Pest number |
|---|---|---|---|
| 1 | 0-1115 | Rice leaf roller | 1115 |
| 2 | 1116-1601 | Rice leaf caterpillar | 485 |
| 3 | 1602-1862 | Paddy stem maggot | 261 |
| 4 | 1863-2915 | Asiatic rice borer | 1053 |
| 5 | 2916-3419 | Yellow rice borer | 504 |
| 6 | 3420-3925 | Rice gall midge | 506 |
| 7 | 3926-4294 | Rice stem fly | 369 |
| 8 | 4293-5132 | Brown plant hopper | 838 |
| 9 | 5133-6021 | White backed plant hopper | 889 |
| 10 | 6022-6574 | Small brown plant hopper | 553 |
| 11 | 6575-7430 | Rice water weevil | 856 |
| 12 | 7431-7834 | Rice leafhopper | 404 |
| 13 | 7835-8007 | Grain spreader thrips | 173 |
| 14 | 8008-8416 | Rice shell pest | 409 |

where $AP(i)$ is the average precision of the $i$-th category and $M$ is the total number of categories.

## 4.4 Results

The proposed model CATransU-Net is a modified U-Net by combining MSU-Net and AU-Net [19]. Fig 5 shows the *mAP* losses of U-Net, MSU-Net, AU-Net and CATransU-Net versus the number of iterations, under the initial values of the training parameters.

From Fig 5, it is found that the losses of all models tend to be stable after 2,000 times, and the loss of U-Net fluctuates greatly, and CATransU-Net is superior to other models, MSU-Net is slightly better than AU-Net. The results verify that the multiscale convolution is suitable for various pest detection, and attention mechanism can improve the detection performance. CATransU-Net has good and fast convergence due to the advantages of multi-scale convolution and attention mechanism.

FFCV is adopted to trained and fine-tune the hyperparameters of CATransU-Net, and the trained model is employed to the test subset to evaluate its performance. Fig 6 shows the detected rice pests by the trained CATransU-Net and 6 comparison models for visual performance comparison: baseline U-Net, TransUnet, DMSAU-Net, HDLU-Net and TinySeg-former. From Fig 6A, it is found that all pests are large and obvious. From Figs 6B–G, it is seen that all models can detect the pests well, and the pests detected by DMSAU-Net are closer to the labeled pests than other baselines. To better visualize the difference between detected pests and labeled pests, key areas are highlighted with appropriate red circles.

To further verify the effectiveness and robustness of CATransU-Net compared with 5 models, Fig 7 shows the pest detection results of 7 rice pest images with complex, fuzzy, small-scale, multi-scale, single pest, multi-pests and complex background respectively. As seen from Fig 7B, it is difficult for U-Net to detect the various complex pests, because the encoder of U-Net does not take into account the diversity of pests, and its max-pooling results in the loss of detailed characteristics of fine pests.

The results from Figs 6 and 7 validate that the proposed method can accurately detect the pests, and effectively eliminate the irrelevant field background and noise in the image. This advantage is crucial in practical pest detection, especially in analyzing the impact of pests in the complex environment of rice fields.

To quantitatively test the detection performance of the proposed model CATransU-Net, Table 2 the average values of the detected results and training time of the six models are shown in Table 2. It is seen from Table 2, CATransU-Net

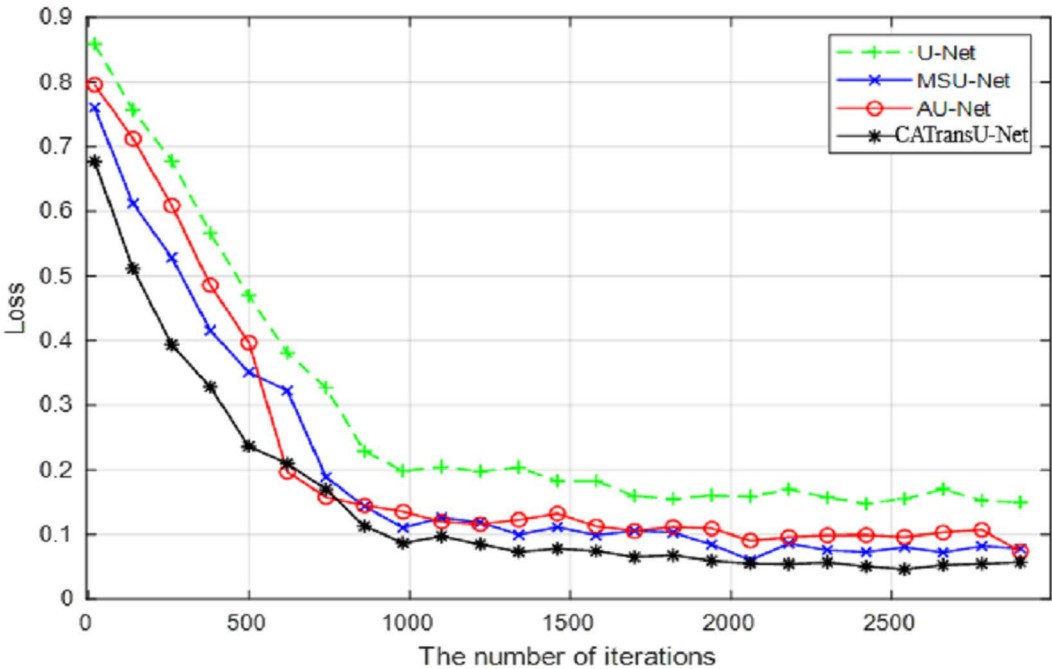

**Fig 5. The losses of *mAP* of four models with the number of iterations.**

is generally superior to other methods, while its training time is slightly long, which is acceptable. The reason is that the processing time of Transformer is long. Although U-Net has less training time, the detected results are low, and the pest details of small defects are lost.

The results in Table 2 validate that CATransU-Net outperforms the other models. Its *Pre* is over 93%, indicating very high correctness in field pest detection task, its *Rec* of 92.24% means CATransU-Net can effectively detect most of the true pest targets, and its *mAP* of 92.62% indicates CATransU-Net can maintain high performance when considering both precision and recall to evaluate the model performance across categories. The main reason is that CATransU-Net can benefit greatly from multiscale convolution, dual Transformer-attention and context attention feature fusion skip-connection, which can learn the global-context and discriminant features for field pest detection. From Table 2, it is seen that the training time of CATransU-Net is 1.64 hour, which is long for fact application. The problem can be mitigated by Transfer learning.

To demonstrate the robustness of CATransU-Net components, a number of FFCV ablation experiments are performed using its different components, where U-Net is regarded as the baseline method for ablation experiments. The *mAP*s are shown in Table 3, where'√' indicates that this module is added into U-Net.

From Table 3, it is seen that adding different modules to U-Net can improve the pest detection *mAP*. Due to the different functions of RDI, DTA and CASC modules, their contributions to pest detection are different. It is also found that RDI, DTA and CASC can greatly improve *mAP* due to the various multiscale field changeable pests.

The results from Table 3 verify the effectiveness of each component of CATransU-Net. To intuitively demonstrate the impact of each component on performance, some visual comparison validation experiments are conducted. The detected pests are shown in Fig 8. As seen from Fig 8, under the same experimental conditions, adding the RDI module to the baseline model U-Net resulted in a greater emphasis on scene details in the pest images, but it also misclassified some details as detected pests. This flaw is significantly improved after the addition of DTA and CASC, respectively, which can improve the accuracy of salient pest detection and reduce background and noise interference. CATransU-Net

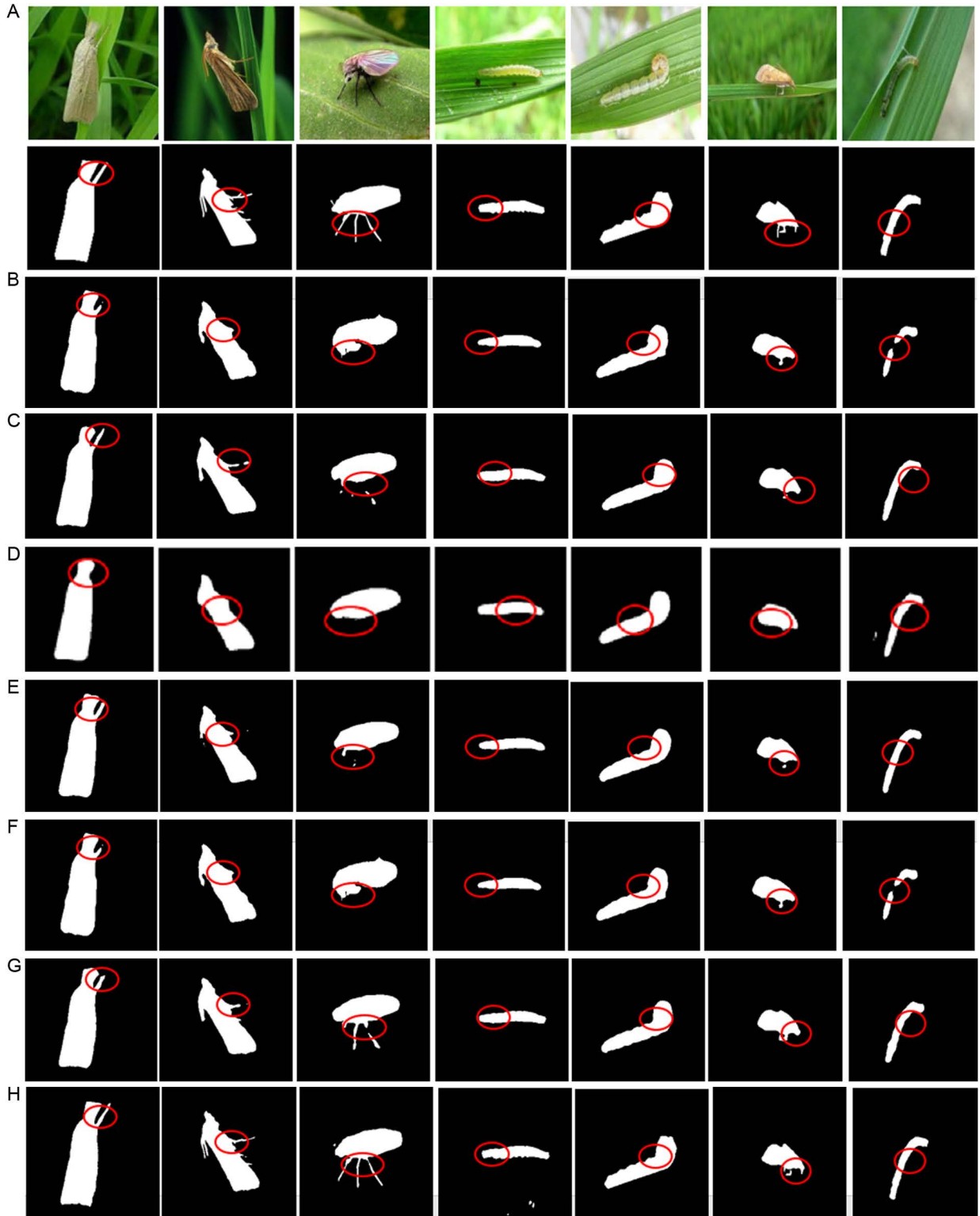

**Fig 6. The detected pest images of obvious pests by 7 methods.** (A) Original simple rice pest images and labeled pest images. (B) U-Net. (C) TransUnet. (D) DMSAU-Net. (E) HDLU-Net. (F) Swin Transformer. (G) TinySegformer. (H) CATransU-Net.

**Table 2. The detected results of 6 methods and their training time.**

| Method Result | U-Net | TransUnet | DMSAU-Net | HDLU-Net | Swin Transformer | TinySegformer | CATransU-Net |
|---|---|---|---|---|---|---|---|
| *Prec* (%) | 84.15 | 91.56 | 85.22 | 85.70 | 90.22 | 91.63 | 93.51 |
| *Rec* (%) | 82.43 | 89.85 | 85.05 | 84.32 | 90.01 | 91.46 | 92.24 |
| *mAR* (%) | 83.96 | 90.70 | 80.58 | 84.21 | 90.11 | 90.38 | 92.62 |
| Training time (h) | 1.01 | 1.92 | 1.45 | 1.61 | 1.91 | 1.87 | 1.64 |

demonstrates a noticeable improvement in detection accuracy after incorporating CASC, enabling accurate detection of field pests. From Fig 8, it is intuitively concluded that the addition of each module enhances the overall performance of the model.

Fig 9 shows the detected pests by adding two components, and the detected results are overall better than that showing in Fig 8.

To rigorously evaluate the trade-offs between performance and computational costs, we compare our proposed model against U-Net variants and recent SOTA pest detection methods in terms of parameters, GFLOPs, and inference time, as shown in Table 4. From Table 4, it is found CATransU-Net outperforms the other models.

To further validate CATransU-Net, a lot of experiments are conducted on the AgriPest dataset. Fig 10 illustrates the detected pest examples. From Fig 10, compared to other models, it is found that the detected pests by CATransU-Net are clear and complete continuous legs and antennae.

## 4.5 Result analysis

From Figs 6–10 and Tables 2–4, it is concluded that CATransU-Net can effectively detect and locate various pests, and outperforms the other models in detection results and training time. The detected pests from Figs 6–10 validate that CATransU-Net can improve the quality of detail detection of field pests. TinySegformer is the second best, TransUnet is better than the other models. From Figs 6, 7 and 10, it is seen that CATransU-Net, TransUnet and TinySegformer can detect the small and vague pests, while DMSAU-Net and HDLU-Net miss some small and vague pest details. A reason is the ability to favor the U-Net architecture, which can reconstruct the original mask image patch from partial observations of the image patch. It is because that combining Transformer with multiscale dilated convolution (MSDC) effectively introduces locality into visual Transformer, where MSDC can extract multiscale features, and Transformer can capture global context features, which can overcome the limitation of the lack of context of the extracted feature blocks in the U-Net encoding path. High-level convolutional blocks are used to encode local information in the early stage, and low-level transform blocks are used to aggregate global information in the late stage. Therefore, by aggregating local-global feature-maps at different layers, the hybrid model can learn rich multiscale discriminant features, and enhance the feature extraction capability. In the CATransU-Net structure, the residual connection, multiscale convolution, dilated convolution, $1 \times 1$ convolution, element-wise concatenation, TSA and GSA are very useful for representing the multiscale, fine-grained global-local context of the various field pests.

From the above experiments, we conclude a lot of explanations for both correct and incorrect detection cases, as follows,

• Correct Detections: Success Cases

   Example 1: Small Pest Localization (Helicoverpa armigera eggs)
   The reason is as follows,
   Context: Eggs (5–10 pixels) on soybean leaves, often obscured by leaf veins.
   Multi-scale DRI Block: Captured fine textures via dilated convolutions (rate = 1).

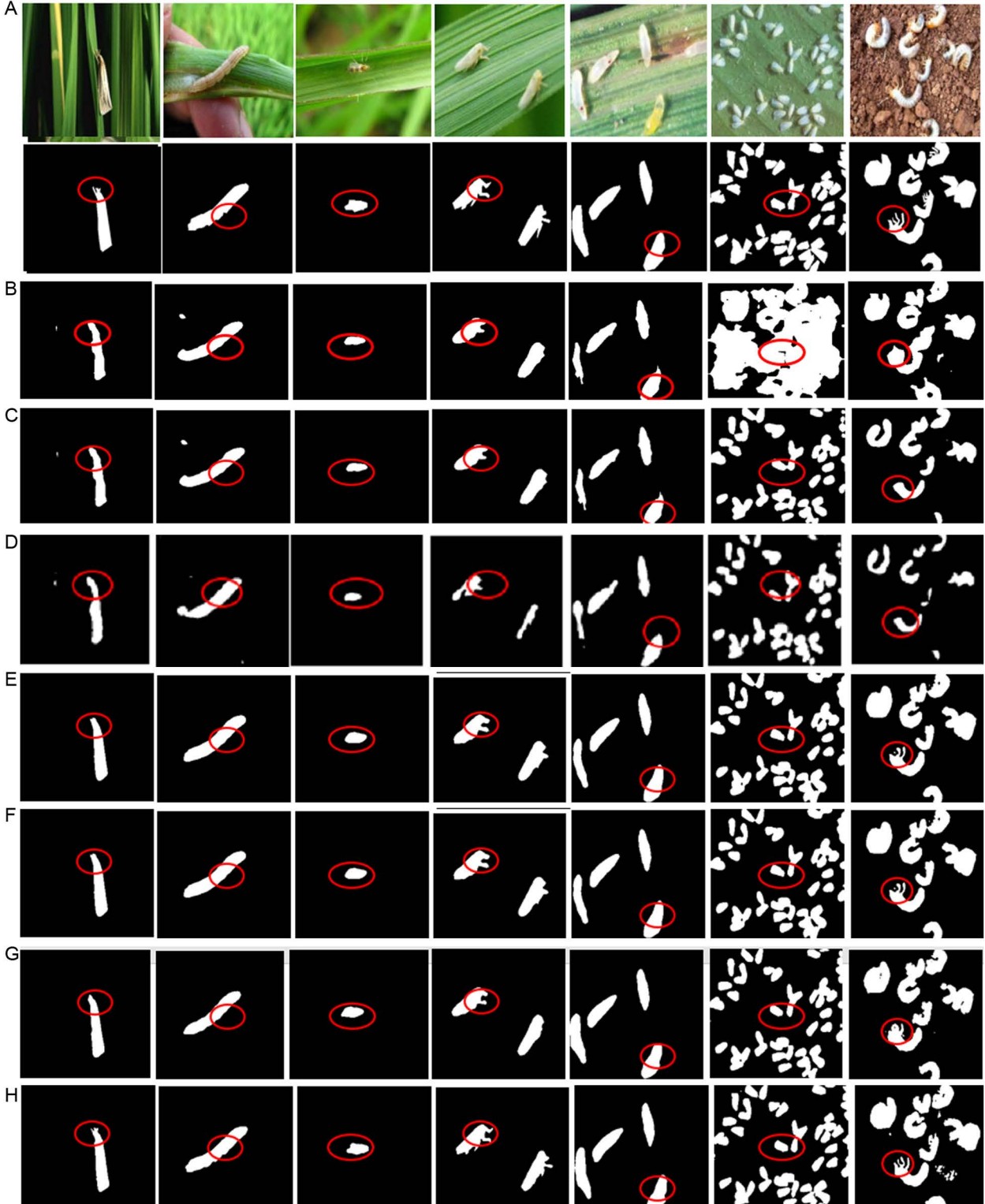

**Fig 7. The detected pest images with complex background by 7 methods.** (A) Original complex rice pest images and labeled pest images. (B) U-Net. (C) TransUnet. (D) DMSAU-Net. (E) HDLU-Net. (F) Swin Transformer. (G) TinySegformer. (H) CATransU-Net.

## Table 3. Pest detection accuracies by adding different modules.

| No. | RDI | DTA | CAFF | mAP(%) |
|---|---|---|---|---|
| 1 | -- | -- | -- | 84.15 |
| 2 | √ | -- | √ | 85.64 |
| 3 | Inception | ASPP | -- | 87.12 |
| 4 | -- | ASPP | √ | 84.50 |
| 5 | √ | √ | -- | 90.17 |
| 6 | -- | √ | √ | 86.23 |
| 7 | √ | ASPP | √ | 91.31 |
| 8 | √ | GSA | √ | 92.10 |
| 9 | √ | TSA | √ | 91.73 |
| 10 | Dilated Inception | √ | √ | 92.34 |
| 11 | √ | √ | √ | 91.62 |

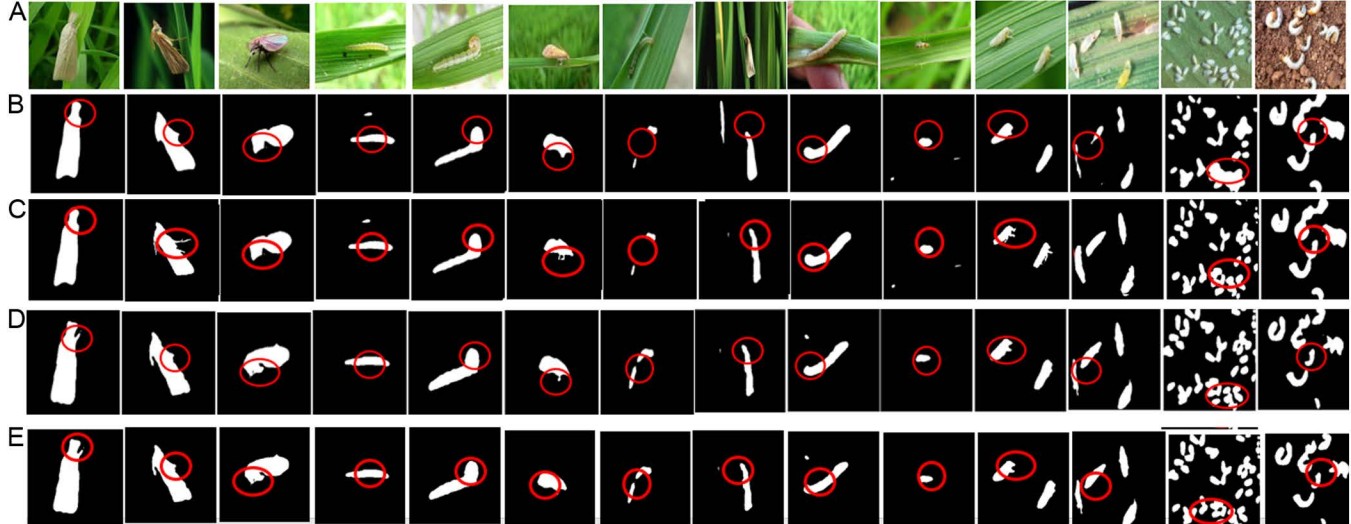

**Fig 8. The comparison of detected pests by adding each component.** (A) Original images. (B) U-Net. (C) U-Net+ RDI. (D) U-Net+ DTA. (E) U-Net+ CASC.

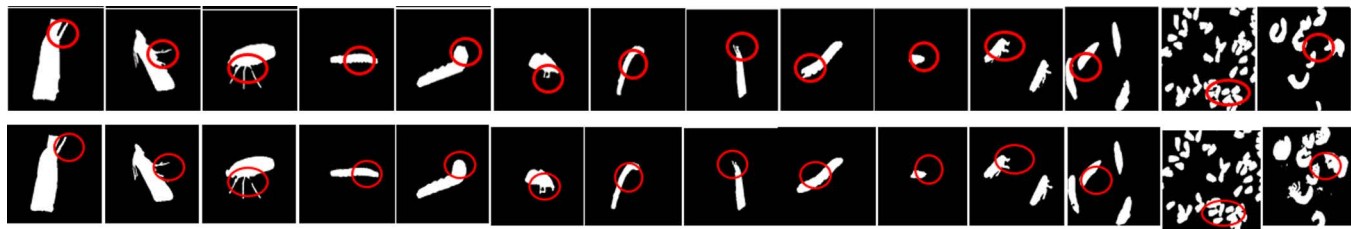

**Fig 9. The comparison of detected pests by adding two components.** (F) U-Net + DTA and CASC. (G) U-Net+ RDI and DTA.

**Table 4. Pest detection set and results.**

| Model | Params (M) | GFLOPs | Inference Time (ms) | Prec.(%) |
|---|---|---|---|---|
| U-Net [19] | 7.8 | 15.2 | 8.2 | 84.15 |
| TransUnet [12] | 36.5 | 42.3 | 22.8 | 91.56 |
| DMSAU-Net [31] | 24.7 | 31.6 | 18.3 | 85.22 |
| HDLU-Net [7] | 18.9 | 26.2 | 15.7 | 85.70 |
| Swin Transformer [42] | 50M | 8.7 | 20.8 | 90.22 |
| TinySegformer [18] | 5.2 | 9.8 | 6.5 | 91.63 |
| **Proposed Model** | 14.6 | 18.9 | 9.4 | 93.51 |

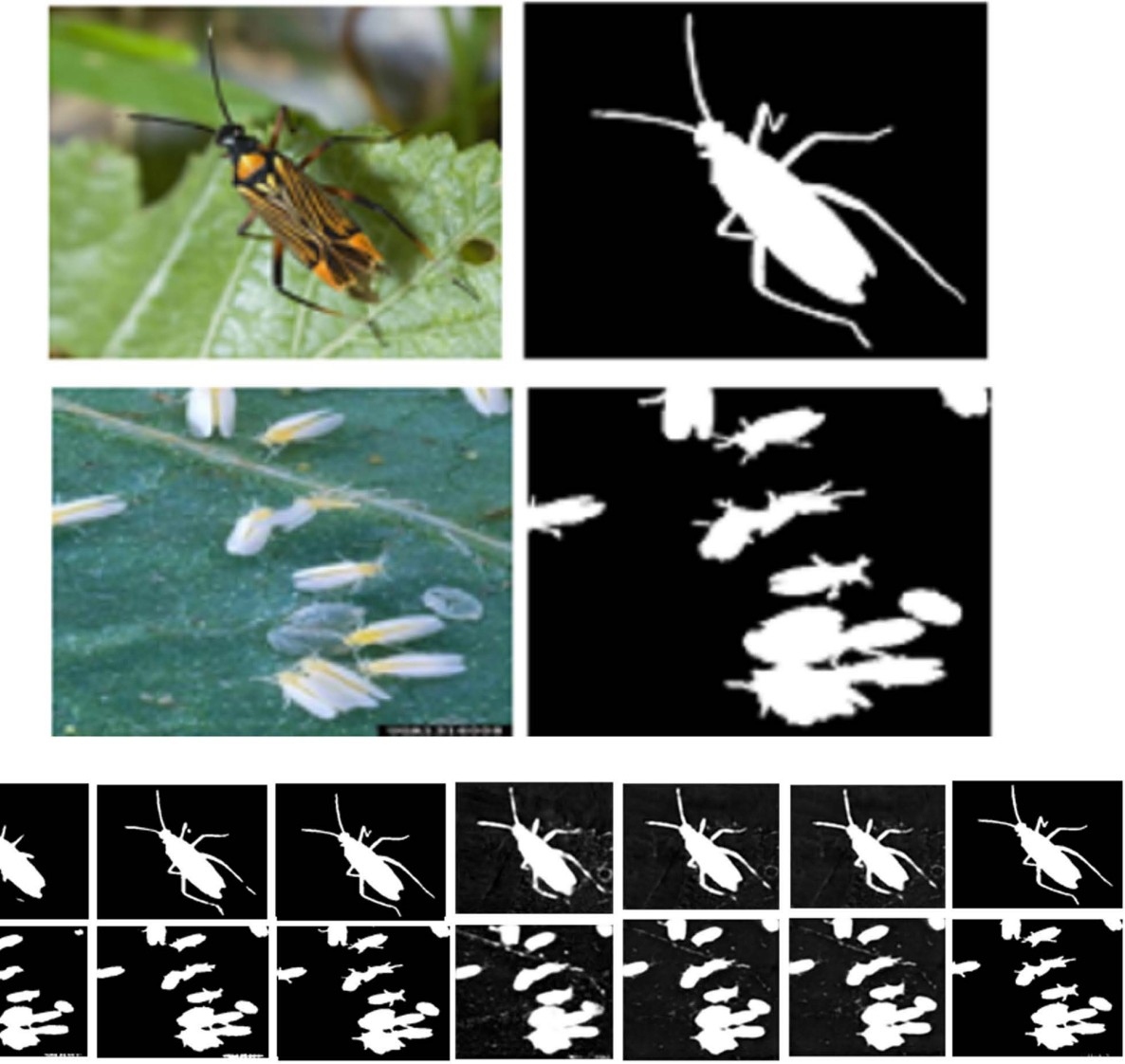

**Fig 10. Visualized detected pest images with complex shapes and background by 7 methods, where (A) Original complex pest images and labeled pest images, (B) U-Net, (C) TransUnet, (D) DMSAU-Net, (E) HDLU-Net, (F) Swin Transformer, (G) TinySegformer and (H) CATransU-Net.**

Cross-Attention: Fused CNN (local edges) and Transformer (global context) to reduce false negatives.

Visual Evidence: See Fig 9a (green boxes).

Example 2: Occluded Pest Clusters (Spodoptera litura larvae)

Context: Dense groups of larvae hidden under leaf debris.

The reason is as follows,

Scale-Adaptive Attention: Dynamically adjusted dilation rates to cover clustered pests.

Residual Links: Preserved spatial details despite occlusion.

- Incorrect Detections: Failure Modes & Fixes

  Example 1: False Positives (Soil Misclassified as Pests)

  The reason is as follows,

  Soil cracks resembled pest textures under low light (Fig 9b, red boxes).

  Solution: Added background-class embeddings in the decoder (Sec. 3.3) to suppress non-pest features.

  Example 2: False Negatives (Transparent Aphids)

  The reason is as follows,

  Near-invisible aphids on glossy leaves (reflectance ≈ background).

  Root Issue: Limited training data for rare "glass-wing" aphid variants.

  Mitigation: Synthetic data augmentation using GAN-generated transparent pests (future work).

From the above analysis, the proposed CATransU-Net can learn discriminant features of the field pest images, and effectively detect field rice pests. In CATransU-Net, dual Transformer-attention can effectively capture the context of the obtained features in encoder, which can provide more global semantic information to the decoder at the lower stage, generating the effective local-global features.

## 5 Conclusion

Detection of pests in paddy field is an important means to ensure rice management, maintain ecological balance and increase rice yield and quality. In this study, Cross-Attention TransU-Net (CATransU-Net) model is constructed to address the limitations of U-Net in various rice pest detection in the field. In CATransU-Net, the convolution module of U-Net is replaced by residual dilated Inception module, the skip-connection is modified by cross-attention skip-connection (CASC) to concatenate the features of encoder and decoder, and dual Transformer-attention module is introduced to the bottleneck of the model to flexibly aggregate contextual feature maps from different semantic scale decoders to generate discriminant feature representations. CATransU-Net is demonstrated on the publicly available rice pest image subset of IP102 with various pests in the complex fields. Due to the unpredictable field environment, such as occlusion, uneven light and mussy background, paddy pests appear in various shapes and sizes in the field. Future work will focus on improving the detection accuracy and model lightweight of CATransU-Net, and measure real-world FPS/power consumption in drone-mounted pest detection scenarios.

## Author contributions

**Conceptualization:** Yunlong Zhang.

**Formal analysis:** Xuwei Lu.

**Funding acquisition:** Yunlong Zhang.

**Investigation:** Yunlong Zhang.

**Methodology:** Xuwei Lu.

**Project administration:** Yunlong Zhang.

**Resources:** Xuwei Lu.

**Software:** Congqi Zhang.

**Validation:** Congqi Zhang.

**Writing – original draft:** Xuwei Lu.

**Writing – review & editing:** Xuwei Lu, Yunlong Zhang, Congqi Zhang.

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
