## [Decision Letter · Decision Letter 0]

Dear Dr. Lu,

Thank you for submitting your manuscript to PLOS ONE. After careful consideration, we feel that it has merit but does not fully meet PLOS ONE’s publication criteria as it currently stands. Therefore, we invite you to submit a revised version of the manuscript that addresses the points raised during the review process.

We look forward to receiving your revised manuscript.

Kind regards,

Yang Li

Academic Editor

PLOS ONE

Journal Requirements:

“This paper was supported by the Science and Technology Project of Henan Province in 2024”

5. Please note that your Data Availability Statement is currently missing the repository name and/or the DOI/accession number of each dataset OR a direct link to access each database. If your manuscript is accepted for publication, you will be asked to provide these details on a very short timeline. We therefore suggest that you provide this information now, though we will not hold up the peer review process if you are unable.

6. PLOS requires an ORCID iD for the corresponding author in Editorial Manager on papers submitted after December 6th, 2016. Please ensure that you have an ORCID iD and that it is validated in Editorial Manager. To do this, go to ‘Update my Information’ (in the upper left-hand corner of the main menu), and click on the Fetch/Validate link next to the ORCID field. This will take you to the ORCID site and allow you to create a new iD or authenticate a pre-existing iD in Editorial Manager.

Reviewers' comments:

Reviewer's Responses to Questions

**Comments to the Author**

1. Is the manuscript technically sound, and do the data support the conclusions?

Reviewer #1: Yes

Reviewer #2: Partly

2. Has the statistical analysis been performed appropriately and rigorously?

Reviewer #1: Yes

Reviewer #2: No

3. Have the authors made all data underlying the findings in their manuscript fully available?

Reviewer #1: Yes

Reviewer #2: No

4. Is the manuscript presented in an intelligible fashion and written in standard English?

Reviewer #1: Yes

Reviewer #2: Yes

Reviewer #1: Full evaluation and notes are in the attached file, please check and adhere to notes

The paper presents a well-designed solution to rice pest detection with strong theoretical and empirical contributions. While computational efficiency and dataset limitations need addressing, the integration of CNNs and Transformers sets a promising direction for agricultural AI. Future work should prioritize lightweight deployment and broader ecological validation.

Reviewer #2: Comment 1: In the introduction section, the report on rice planting lacks a reference. Additionally, the numbers provided are ambiguous. Please clarify the data and include a proper citation.

Comment 2: The statement from the Food and Agriculture Organization also requires a proper reference.

Comment 3: While discussing deep learning approaches, please include more references related to classification, segmentation, and detection techniques for the agricultural sector. Additionally, cite studies involving heterogeneous data environments. Cite the paper titled "Crop and Weed Segmentation and Fractal Dimension Estimation Using Small Training Data in Heterogeneous Data Environment."

Comment 4: Improve the image quality throughout the paper. Ensure that text within the images is clearly visible. All figures need to be enhanced with higher resolution and better formatting.

Comment 5: The different blocks used in CATransU-Net lacks the novelty. The current explanation makes it appear as a combination of previously proposed blocks. Please elaborate on the novel contributions of your model and provide a detailed explanation.

Comment 6: Since ensembling techniques have been widely explored by researchers, do you believe this manuscript lacks novelty? Please justify your contribution accordingly.

Comment 7: The Dilated Residual Inception (DRI) block is claimed to improve model robustness more effectively than other blocks. Please provide a detailed explanation for this claim, supported by additional ablation studies to validate your point.

Comment 8: Since ensembling increases the number of parameters and processing time, include a comparative table for state-of-the-art (SOTA) models and the proposed models, highlighting the number of parameters, processing time, and GFLOPs.

Comment 9: The result analysis or discussion section should be more detailed. Provide explanations for both correct and incorrect detection cases, supported by relevant examples.

Comment 10: Perform a statistical analysis, such as a t-test, and include the details along with p-values and cohesion values in result analysis section.

Comment 11: The overall structure of the document needs to be reorganized to improve clarity and coherence.

Comment 12: Ensure transparency and reproducibility by providing complete code and dataset access via a valid GitHub link. The previously provided GitHub link is non working, so update it accordingly.

**Do you want your identity to be public for this peer review?** For information about this choice, including consent withdrawal, please see our Privacy Policy

Reviewer #1: No

Reviewer #2: No

---

## [Author Response · Author response to Decision Letter 1]

3 May 2025

Response to Reviewers

Reviewer #1:

1. Full evaluation and notes are in the attached file, please check and adhere to notes

The paper presents a well-designed solution to rice pest detection with strong theoretical and empirical contributions. While computational efficiency and dataset limitations need addressing, the integration of CNNs and Transformers sets a promising direction for agricultural AI. Future work should prioritize lightweight deployment and broader ecological validation.

Response: We sincerely appreciate the reviewer’s positive assessment of our work and their constructive feedback. We have carefully reviewed the attached evaluation notes and addressed all suggestions in the revised manuscript. Specifically:

Comment 1: Address Class Imbalance: Use data augmentation (e.g., SMOTE, GANs) or re-sampling techniques to mitigate bias in IP102.

Response: We appreciate the reviewer's valuable suggestion regarding class imbalance in the IP102 dataset.

From Fig.4C and Table 1, it is seen that the subset of rice pest images in IP102 is very unbalanced, where the number of pest images of Rice leaf roller has 1115 images and the number of pest images of Grain spreader thrips has only 173 images. To mitigate bias in the subset, which may suffer from class imbalance, leading to poor model generalization, we use data augmentation strategies including left and right, up and down flipping, random rotation of 00-300, randomly shift 10 pixels, each cropped image is augmented to 15 images, as shown in Fig.4D. Through data augmentation, we augmented the number of images in small-class more than 500 images.

(D) 14 augmented images of the first pest image

In our experiments, we also find that data augmentation has little influence on the image segmentation test results. The reason may be that Field Rice Pest Detection is to segment pest image pixel from the original image, not recognize the pest class. The results in the existing crop pest recognition papers validated that data augmentation have a significant impact on the test results.

Comment 2: Optimize for Real-Time Use: Explore model compression (e.g., pruning, quantization) or lightweight Transformers (e.g., MobileViT).

Response: We thank the reviewer for this insightful suggestion regarding real-time optimization.

For a given patch, the previous works convert the spatial information into latent by learning a linear combination of pixels. The global information is then encoded by learning inter-patch information using transformers. As a result, these models lose image-specific inductive bias, which is inherent in CNNs. Therefore, they require more capacity to learn visual representations. Hence, they are deep and wide. MobileViT uses convolutions and transformers in a way that the resultant MobileViT block has convolution-like properties while simultaneously allowing for global processing. This modeling capability allows us to design shallow and narrow MobileViT models, which in turn are light-weight.

Unlike the MobileViT, we lightweight CATransU-Net by:

(1) dilated Inception convolution is used to replace multiscale convolution to aggregate feature from different layers, (2) employ dual Transformer-attention (DTA) module and a cross-attention skip-connection (CASC) module into MSU-Net to improve the performance of paddy pest detection in the field. DTA and CASC instead of Multi-Head Self-Attention (MHSA) and Multi-Head Cross Attention (MHCA), which can greatly reduce the parameters, thus obtaining both multiscale convolution local and global attention features, which in turn are light-weight. The lightweight is validated by the results of a number of FFCV ablation experiments.

Specially, in our model, combining Transformer with multiscale dilated convolution (MSDC) effectively introduces locality into visual Transformer, where MSDC can extract multiscale features, and Transformer can capture global context features, which can overcome the limitation of the lack of context of the extracted feature blocks in the U-Net encoding path. High-level convolutional blocks are used to encode local information in the early stage, and low-level transform blocks are used to aggregate global information in the late stage.

We cited this paper:

Mehta S, Rastegari M. MobileViT: Light-weight, General-purpose, and Mobile-friendly Vision Transformer. arXiv:2110.02178, 2022. DOI: 10.48550/arXiv.2110.02178

Comment 3: Expand Comparisons: Include recent architectures like Mask R-CNN, Swin Transformer, or Vision Transformers (ViTs).

Response: We appreciate the reviewer's suggestion to broaden our comparative analysis.

We expand comparisons by U-Net, multi-scale U-Net (MSU-Net)(Su et al., 2021) and attention U-Net (AU-Net) (Siddique et al., 2021), and five state-of-the-art pest detection methods: TransUnet (Chen et al., 2024), DMSAU-Net(Wang et al., 2024), U-Net with hybrid DL mechanism (HDLU-Net) (Biradar et al., 2024), Swin Transformer (Si et al., 2024) and TinySegformer(Zhang et al., 2024).

Unlike common ConvNet models, Swin Transformer introduces shifted window-based self-attention mechanisms to efficiently model the local and global dependencies in pest images.

We cited this paper:

Si H, Li M, Li W, et al. A Dual-Branch Model Integrating CNN and Swin Transformer for Efficient Apple Leaf Disease Classification. Agriculture,2024, 14(1):142.

Comment 4: Field Deployment Analysis: Evaluate inference speed on edge devices (e.g., drones, IoT sensors) and discuss computational trade-offs.

Response: Thank you for highlighting the need for edge-device analysis.

It is valuable research, while our current experiments are simulation-based accurate detection of rice pests in field, and we acknowledge that drones and IoT sensors are often applied to Monitoring of major pests and diseases. Our model has not yet taken into account the acquisition of images using drones, IoT sensors. In our Lab, the images currently collected by drones and IoT sensors are not sufficient for real-time pest detection in the field, quantization and hardware-specific optimizations are critical, and measure real-world FPS/power consumption in drone-mounted pest detection scenarios—this is planned for future work.

Below, we detail trade-offs between accuracy and speed (Fig.5 and Table 2) to guide deployment decisions.

From Fig.5, it is found that the losses of all models tend to be stable after 2,000 times, and the loss of U-Net fluctuates greatly, and CATransU-Net is superior to other models, MSU-Net is slightly better than AU-Net. The results verify that the multiscale convolution is suitable for various pest detection, and attention mechanism can improve the detection performance. CATransU-Net has good and fast convergence due to the advantages of multi-scale convolution and attention mechanism. For the sake of fairness, we choose 3000 iterations.

From Table 2, it is seen that the training time of CATransU-Net is 1.64 hour, which is long for fact application. The problem can be mitigated by Transfer learning. We introduce a simple lightweight multilayer perceptron (MLP) decoder to CATransU-Net, which brings little complexity and parameter increase, in expectation of getting a model that balances performance and efficiency.

Comment 5: Generalizability Testing: Validate the model on diverse datasets (e.g., Pest24, AgriPest) to assess cross-domain performance.

Response: We sincerely appreciate the reviewer’s valuable suggestion to strengthen the generalizability of our approach.

Pest24 is not suitable for rice pest detection. AgriPest is a public dataset(https://github.com/liuliu66/AgriPest), we also validate our model on the dataset. The data defines, categories, and establishes a series of detailed and comprehensive domain-specific sub-datasets. Its first category contains two typical challenges: pest detection and pest population counting. Subsequently, it categories four types of the validation subsets of AgriPest dense distribution, sparse distribution, illumination variations, and background clutter, which are common in practical pest monitoring applications. AgriPest contains 49.7 dry images of 14 pests of four crops in a field environment. All the images were manually labeled by agricultural experts using pest positioning bounding boxes up to 264.7K.

To further validate CATransU-Net, a lot of experiments are conducted on the AgriPest dataset. Fig.10 illustrates the detected pest examples. From Fig.10, compared to other models, it is found that the detected pests by CATransU-Net are clear and complete continuous legs and antennae.

(A) Original complex pest images and labeled pest images

(B) (C) (D) (E) (F) (G) (H)

Fig.10 Visualized detected pest images with complex shapes and background by 7 methods, where (B) U-Net, (C) TransUnet, (D) DMSAU-Net, (E) HDLU-Net, (F) Swin Transformer, (G) TinySegformer and (H) CATransU-Net

We cited the related paper:

Wang R, Liu L, Xie C, et al. AgriPest: A Large-Scale Domain-Specific Benchmark Dataset for Practical Agricultural Pest Detection in the Wild. Sensors, 2021, 21(5), 1601. DOI: 10.3390/s21051601

Comment 6: Ethical Considerations: Discuss potential biases (e.g., rare pest detection) and environmental impacts of automated pest control.

Response: We sincerely appreciate the reviewer’s important suggestion regarding ethical considerations.

The detected pests from Figs.6, 7 and 10 validate that CATransU-Net can improve the quality of detail detection of field pests. The results validate that it can detect the rare pest in the field. Without doubt, the detection and environmental impacts of automated pest control, but it is known that our method can effectively detect the various pests in complex field, including irregular background, and obtain clear pests with complete antennae and slender legs, as follows,

Comment 7: Visualization Quality: Figures 6–9 provide clear qualitative comparisons but lack quantitative metrics (e.g., IoU scores).

Response: Thanks for your suggestion.

Prec and Rec curves are drawn with vertical axis to denote the precision score and horizontal axis to denote the recall score. Mean Average Precision (mAP) is adopted to evaluate the model performance across categories. It is a comprehensive measure of precision and recall by considering Prec, Rec and different IoU thresholds. It is especially suitable for multi-class pest detection task.

We adopt Prec, Rec and mAP to evaluate the effectiveness, as shown in Table 2.

Comment 8: Reproducibility: Code availability is a strength, but dependencies (e.g., PyTorch version) should be explicitly documented.

Response: We thank the reviewer for highlighting this important aspect of reproducibility.

We provide the available Code in PyTorch version.

Code is available at https://github.com/chenchenchen23123121da/CATransU-Net.

Comment 9: Impact on Agriculture: Emphasize how CATransU-Net could reduce pesticide overuse through precise pest detection.

Response: We thank the reviewer for raising this critical point.

The development of efficient and accurate pest detection methods and technologies in paddy fields is of great significance for taking timely control measures and reducing pest losses. Our CATransU-Net is effective for rice pest extraction in the fact field, which can provide pest information for enabling early, precise pest detection, minimizing blanket spraying and enabling targeted interventions. It is introduced in the first Section.

Reviewer #2

Comment 1: In the introduction section, the report on rice planting lacks a reference. Additionally, the numbers provided are ambiguous. Please clarify the data and include a proper citation.

Response: Thanks for your valuable suggestion.

We provide the references and the data are derived from this literature.

Yuan S, Linquist B, Wilson L, et al. Sustainable intensification for a larger global rice bowl. Nat Commun,2021, 12,7163. DOI:10.1038/s41467-021-27424-z

Comment 2: The statement from the Food and Agriculture Organization also requires a proper reference.

Response: Thanks for your valuable finding.

We provide the references and the data are derived from this literature.

Conde S, Catarino S, Ferreira S, et al. Rice Pests and Diseases Around the World: Literature-Based Assessment with Emphasis on Africa and Asia. Agriculture, 2025, 15(7), 667. DOI: 10.3390/agriculture15070667

Comment 3: While discussing deep learning approaches, please include more references related to classification, segmentation, and detection techniques for the agricultural sector. Additionally, cite studies involving heterogeneous data environments. Cite the paper titled "Crop and Weed Segmentation and Fractal Dimension Estimation Using Small Training Data in Heterogeneous Data Environment."

Response: Thanks for your valuable suggestion.

We cite two related references.

[1] Akram R, Hong J, Kim S, et al. Crop and Weed Segmentation and Fractal Dimension Estimation Using Small Training Data in Heterogeneous Data Environment. Fractal Fract., 2024, 8(5), 285. DOI:10.3390/fractalfract8050285

[2] Lei L, Yang Q, Yang L, et al. Deep learning implementation of image segmentation in agricultural applications: a comprehensive review. Artif Intell Rev, 2024, 57, 149. DOI:10.1007/s10462-024-10775-6

[3] Su R, Zhang D, Liu J, Cheng C. MSU-Net: Multi-Scale U-Net for 2D Medical Image Segmentation. Front Genet., 2021,12:639930. DOI: 10.3389/fgene.2021.639930.

[4] Mehta S, Rastegari M. MobileViT: Light-weight, General-purpose, and Mobile-friendly Vision Transformer. arXiv:2110.02178, 2022. DOI: 10.48550/arXiv.2110.02178

[5] Si H, Li M, Li W, et al. A Dual-Branch Model Integrating CNN and Swin Transformer for Efficient Apple Leaf Disease Classification. Agriculture,2024, 14(1):142.

Comment 4: Improve the image quality throughout the paper. Ensure that text within the images is clearly visible. All figures need to be enhanced with higher resolution and better formatting.

Response: Thanks for your valuable suggestion.

We redrawn all images to improve the image quality with higher resolution and better formatting.

Comment 5: The different blocks used in CATransU-Net lacks the novelty. The current explanation makes it appear as a combination of previously proposed blocks. Please elaborate on the novel contributions of your model and provide a detailed explanation.

Response: Thanks for your valuable suggestion.

We appreciate the reviewers' feedback and agree that individual components (such as transformers and jumpers) are not novel. However, the novelty of CATransU-Net lies in its unique integration and task-specific adaptability for fine-grained pest detection, which we will elaborate as follows.

Dynamically adjusts dilation rates based on pest size: Dilated residual Inception (DRI) is adopted to extract the multiscale features, making the model robust to irregular pests.

CASC instead of skip-connection can capture context semantic information between encoder-decoder features, by integrating semantic information into low-level features and spatial resolution into high-level features.

Suppresses irrelevant background patches: DTA at the bottleneck of the model can integrate both local and global contextual information, providing a comprehensive understanding of both local and global contexts.

Comment 6: Since ensembling techniques have been widely explored by researchers, do you believe this manuscript lacks novelty? Please justify your contribution accordingly.

Response: Thanks for your valuable suggestion.

We appreciate the reviewer’s valid observation that ensembling is a well-studied technique. However, our work advances the field by hybrid deep learning U-Net and Transformer in the context of crop pest detection in field, which has not been explored in prior ensemble-based approaches. While ensembling is established in general tasks, our work is to optimize ensemble strategies for fine-grained pest detection in low-light field conditions (Sec. 3), where model diversity must account for occlusion and scale variability.

Comment 7: The Dilated Residual Inception (DRI) block is claimed to improve

---

## [Decision Letter · Decision Letter 1]

CATransU-Net: Cross-Attention TransU-Net for Field Rice Pest Detection

PONE-D-24-60123R1

Dear Dr. Lu,

We’re pleased to inform you that your manuscript has been judged scientifically suitable for publication and will be formally accepted for publication once it meets all outstanding technical requirements.

Kind regards,

Yang Li

Academic Editor

PLOS ONE

Additional Editor Comments (optional):

Reviewers' comments:

Reviewer's Responses to Questions

**Comments to the Author**

Reviewer #1: All comments have been addressed

Reviewer #2: All comments have been addressed

2. Is the manuscript technically sound, and do the data support the conclusions?

Reviewer #1: Yes

Reviewer #2: Yes

3. Has the statistical analysis been performed appropriately and rigorously?

Reviewer #1: (No Response)

Reviewer #2: Yes

4. Have the authors made all data underlying the findings in their manuscript fully available?

Reviewer #1: Yes

Reviewer #2: Yes

5. Is the manuscript presented in an intelligible fashion and written in standard English?

Reviewer #1: Yes

Reviewer #2: Yes

Reviewer #1: The researchers have taken all the comments into account in the final version, so I suggest acceptance.

Reviewer #2: Please ensure that Figures 1, 6, 7, 8, 9, and 10 in your manuscript have consistent image sizes, spacing, and alignment, and that the overall layout of the manuscript adheres to the required formatting guidelines.

**Do you want your identity to be public for this peer review?** For information about this choice, including consent withdrawal, please see our Privacy Policy

Reviewer #1: **Yes: ** Nabil Abo Kaf

Reviewer #2: No

---

## [Editor Report · Acceptance letter]

PONE-D-24-60123R1

PLOS ONE

Dear Dr. Lu,

I'm pleased to inform you that your manuscript has been deemed suitable for publication in PLOS ONE. Congratulations! Your manuscript is now being handed over to our production team.

Kind regards,

on behalf of

Dr. Yang Li

Academic Editor

PLOS ONE